# How improvements to drug effectiveness impact mass drug administration for control and elimination of schistosomiasis

John R. Ellis[1]*, Nyamai Mutono[2,3], Andreia Vasconcelos[4,5], Samuel M. Thumbi[2,3,6], T. Déirdre Hollingsworth[4], Roy M. Anderson[1]

1 Department of Infectious Disease Epidemiology, School of Public Health, Faculty of Medicine, White City Campus, Imperial College London, London, United Kingdom, 2 Centre for Epidemiological Modelling and Analysis, University of Nairobi, Nairobi, Kenya, 3 Paul G. Allen School for Global Health, Washington State University, Pullman, Washington, United States of America, 4 Big Data Institute, Li Ka Shing Centre for Health Information and Discovery, University of Oxford, Oxford, United Kingdom, 5 Centre for Global Health Research, Nuffield Department of Medicine, University of Oxford, Old Road Campus, Oxford, United Kingdom, 6 Institute of Immunology and Infection Research, University of Edinburgh, Edinburgh, United Kingdom

* john.ellis@imperial.ac.uk

## Abstract

Schistosomiasis affects more than 230 million people worldwide. Control and elimination of this parasitic infection is based on mass drug administration of praziquantel (PZQ), which has been in use for several decades. Because of the limitations of the efficacy of PZQ especially against juvenile worms, and the threat of the emergence of resistance, there is a need to consider alternative formulations or delivery methods, or new drugs that could be more efficacious. We use an individual-based stochastic model of parasite transmission to investigate the effects of possible improvements to drug efficacy. We consider an increase in efficacy compared to PZQ, as well as additional efficacy against the juvenile life stage of schistosome parasites in the human host, and a slow-release formulation that would provide long-lasting efficacy for a period of time following treatment. Analyses suggest a drug with a high efficacy of 99%, or with efficacy lasting 24 weeks after treatment, are the two most effective individual improvements to the drug profile of PZQ. A drug with long lasting efficacy is most beneficial when MDA coverage is low. However, when prevalence of infection has already been reduced to a low level, a high efficacy is the most important factor to accelerate interruption of transmission. Our results indicate that increased efficacy against juvenile worms can only result in modest benefits, but the development of a new drug formulation with higher efficacy against adult worms or long-lasting efficacy would create an improvement to the community impact over the currently used formulation.

**Data availability statement:** The code used for model implementation, simulations, and preparation of figures, is publicly available on GitHub: https://github.com/DrJREllis/SCH-drug-improvements

**Funding:** JE, NM, AV, ST, DH and RA acknowledge funding by the Gates Foundation grant to the Neglected Tropical Diseases Modelling Consortium (INV-030046). The funders had no role in study design, data collection and analysis, decision to publish, or preparation of the manuscript.

**Competing interests:** The authors have declared that no competing interests exist.

## Author summary

The World Health Organization has set the target of elimination of schistosomiasis as a public health problem by 2030. Currently, praziquantel is the sole drug used in mass drug administration (MDA) for schistosomiasis, raising concerns about the potential evolution of resistance and whether an improved drug profile would be necessary to achieve elimination and the more ambitious goal of interruption of transmission. We present the results from an individual-based stochastic mathematical model that simulates schistosome transmission and the impact of MDA. Three potential improvements to the properties of the drug are considered: improved efficacy, long lasting efficacy post a single treatment and improved efficacy against the juvenile life stage of the schistosome. Our findings reveal that with good coverage, an improved drug efficacy is best at reducing prevalence and achieving interruption of transmission. However, when MDA quality is compromised due to low coverage, infrequent treatment or high rates of non-adherence, then a long-lasting efficacy performs best. There is only a modest benefit of improved drug efficacy against juvenile schistosomes. These results highlight the importance of providing frequent treatment at high coverage levels in the population and inform future drug development aims.

## Introduction

The human schistosome parasites are parasitic flatworms that cause schistosomiasis, a disease that affects more than 230 million people worldwide [1]. There are three main species responsible for human infection: *Schistosoma mansoni*, *Schistosoma haematobium* and *Schistosoma japonicum*. Each species cover different geographic areas, *S. mansoni* is endemic in several countries in Africa and the Caribbean, *S. haematobium* is endemic in Africa and Middle East, and *S. japonicum* is endemic in Asia [2]. Like several other neglected tropical diseases (NTDs), mass drug administration (MDA) is a key component in the control and elimination programmes of schistosomiasis. Currently, praziquantel (PZQ) is the only drug available for use against schistosomes and has been in use for over 40 years [3].

Because of the high disease burden, schistosomiasis has been targeted for elimination as a public health problem by the World Health Organization in several countries by 2030 [4]. However, attaining this goal will likely require very high levels of coverage of MDA [5], as well as improvements in access to safe water, sanitation and hygiene (WASH) and possibly some form of snail control [6]. As there is the potential for *Schistosoma* species to develop resistance against PZQ, which also has other limitations such as a low efficacy against the juvenile stage in the schistosome life cycle, alternative drugs have been considered [7,8]. An ideal drug would be efficacious against all stages of the life cycle and include other benefits such as a slow-release mechanism that could act to provide a longer duration of protection post treatment.

The schistosome life cycle is complex, going through multiple development stages [9,10]. When a free-living cercaria infects a human host by penetrating the skin, it goes on a journey through the body, via the vascular system. Juvenile worms travel to the lungs and the liver, before reaching the intestine (*S. mansoni* and *S. japonicum*) or bladder (*S. haematobium*) where they typically remain for 5–10 years as adult worms [11].

Detailed examination of the life cycle within a human host is difficult but estimates of the length of time between infection and egg production in mice hosts has been estimated to be 36–38 days [12]. In that time, the proportion of juvenile worms surviving to maturity has been found to be approximately 40% [9,13]. Studies of the human immune response have found a similar time span of 4–6 weeks between infection and detection of antibodies targeting adult worms [14].

PZQ has been found to be ineffective against juvenile worms, possibly due to their location within the body at the time of treatment [7,15,16]. For example, one study found that the concentration of PZQ required to reduce the worm burden by 50% was 30 times higher when treating after 4 weeks, compared to 7 weeks [13]. Alternatives to PZQ have been suggested for their efficacy against juveniles [7,17], however the impact that this would have at a population transmission level is unknown.

An alternative approach to targeting juveniles at the time of treatment, is to instead administer a slow-release drug, which will continue to kill adult worms for several weeks post pill swallowing. Research has shown that the slow-release delivery of drugs is effective against schistosomes [18,19]. If such a treatment were applied during MDA, it could theoretically target juvenile worms as they mature. If the period of slow release is sufficiently long (longer than the typical maturation period of juveniles), or if the new drug was efficacious against juveniles as well, there is also the added benefit of giving lasting protection against new infections for a period after treatment.

Intuition suggests that coverage and efficacy are the two most important factors in an MDA program and the product of the two will give a good reflection on its effectiveness. However, intuition is less reliable when additional parameters are included, such as the length of time that efficacy will last after treatment. In cases such as this, mathematical models are valuable tools for helping to formulating policy and in drug design, by facilitating quantitative analyses of the impact of different improvements at the population level in the drug profile of action on the parasite.

In this paper, we investigate the impact of drug profile improvements to the current practice of using PZQ in MDA, employing an individual based stochastic mathematical model of the transmission and mass drug treatment of *S. mansoni* and *S. haematobium*. We assess via simulations additional efficacy against juvenile worms as well as adult worms, long lasting efficacy that gives protection against adult worms for several weeks post treatment, and an improvement in drug efficacy against adult worms only. The scale of the improvements to the drug profile are chosen to demonstrate clearly the value of each mechanism of improvement and are not based on any current product in development. To measure the effectiveness of each of these improvements, we show the average time taken to reach interruption of transmission based on 200 repeated simulations, i.e., zero prevalence in the entire population.

## Methods

### Model formulation

We use a stochastic individual based model developed at Imperial College London (the Imperial model) that is based on the deterministic model that has been described in previous publications [20,21]. To include the juvenile stage in a human we adjust the model of mean adult worm burden as a function of age and time, $M(a, t)$, to include the mean juvenile worm burden $J(a, t)$, so that in the deterministic version,

$$\frac{\partial J(a, t)}{\partial t} + \frac{\partial J(a, t)}{\partial a} = L\beta(a) - \mu J(a, t) - \sigma_J J(a, t),$$

$$\frac{\partial M(a,t)}{\partial t} + \frac{\partial M(a,t)}{\partial a} = \mu J(a,t) - \sigma_M M(a,t),$$

where L represents the concentration of the infectious material in the environment, $\beta(a)$ is the age-related contact rate, $\mu$ is the maturation rate of juvenile worms, and $\sigma_J$ and $\sigma_M$ are the natural death rates of juvenile and adult worms respectively.

In the Imperial model, infection and maturation of juvenile worms and death of juvenile and adult worms are discrete events, modelled as a Poisson process. All other details are as described in previous publications [22–24]; we assume the worms are monogamous and have a negative binomial distribution which arises from a gamma distributed predisposition to infection amongst the individual human hosts. This leads to heterogeneity of infection rates within age groups, while individuals also have an age-dependent contact rate with infectious material, and the same age-dependence is also included in the output of eggs, which is density-dependent [25].

We also replicate some of the simulations with the SCHISTOX model, developed at the University of Oxford [26,27]. These two models are similar but differ in how egg production is modelled: SCHISTOX assumes that the number of eggs produced is proportional to the number of worm pairs, whereas the Imperial model assumes egg production is density-dependent so that the higher the burden of infection, the less productive each individual worm is [22]. These models have been previously parameterised and validated using data from a number of sources [20,22,28], while the contact rates and aggregation parameters can be changed to fit to age-dependent infection intensity and prevalence data in particular locations [21]. The two models have been calibrated to have the same prevalence at baseline by varying $R_0$ in the Imperial model and the contact rate in the SCHISTOX model. The parameter values used in model simulations are documented in Table 1 for *S. mansoni* and in Table 2 for *S. haematobium.*

## MDA treatment

Starting in a closed, untreated population, we simulate biannual MDA, with a coverage of 75% in the community (everyone over the age of 2). We assume that treatment is delivered at random at each round. We first consider a 'baseline' approach, whereby MDA of PZQ occurs with an efficacy of 86.3% against *S. mansoni*, or 94% against *S. haematobium*

**Table 1. Parameter values for Schistosoma mansoni.**

| Parameter | Value | Reference |
|---|---|---|
| Fecundity (eggs/female/sample) | 0.34 | [32–34] |
| Aggregation parameter | 0.04-0.24 | [21,28] |
| Density dependent fecundity | 0.0007 | [21,35] |
| Adult worm life span (years) | 5.7 | [11,21,32] |
| Juvenile worm life span (weeks) | 5 | [12] |
| Juvenile survival rate | 0.4 | [9,13] |
| Low adult burden setting: age specific contact rates for 0–5, 5–10, 10–16, 16 + years old | 0.01, 1.2, 1, 0.02 | [35,36] |
| High adult burden setting: age specific contact rates for 0–5, 5–12, 12–20, 20 + years old | 0.01, 0.61, 1, 0.12 | [35,36] |
| Drug efficacy | 86.3% | [29] |
| Aggregation of diagnostic | 0.87 | [33,34,37] |
| Basic reproduction number (Imperial model) | 1.2-4 | – |
| Contact rate (SCHISTOX model) | 0.34 – 2 | – |

**Table 2. Parameter values for Schistosoma haematobium.**

| Parameter | Value | Reference |
|---|---|---|
| Fecundity (eggs/female/10 ml sample) | 3.6 | [38,39] |
| Aggregation parameter | 0.04-0.24 | [28] |
| Density dependent fecundity | 0.0006 | [21] |
| Adult worm life span (years) | 4 | [20,40] |
| Juvenile worm life span (weeks) | 5 | [12] |
| Juvenile survival rate | 0.4 | [9,13] |
| Age specific contact rates for 0–5, 5–10, 10+ years old | 0.3, 1, 0.02 | [24] |
| Drug efficacy | 94% | [29] |
| Aggregation of diagnostic | 0.5 | [28] |
| Basic reproduction number (Imperial model) | 1.2-2 | – |
| Contact rate (SCHISTOX model) | 0.34 – 0.5 | |

[29]. The effect of the drug is assumed to be immediate, so that any adult worm in a host that is treated will have a probability of dying at the time of treatment.

When considering improvements to treatment that include efficacy against juvenile worms, they are also killed at the time of treatment, with the same efficacy as for adult worms. Thus, we do not consider the pharmacokinetics or pharmacodynamics that might change how a drug will react to juvenile or adult worms, only the resulting probability of a worm being killed. In the slow-release treatment, we assume that MDA still occurs as normal, but there is a fixed period of time following treatment whereby a juvenile worm that reaches maturation will die with the same probability as the original drug efficacy. We consider efficacious periods of 8, 16 and 24 weeks. When there is a slow-release treatment that also has efficacy against juveniles, the worm will be killed at the time of infection, rather than the time of maturation.

Initially we investigate the impact of improving drug efficacy to 99%, with different levels of coverage and at different baseline prevalences ($R_0$ values) and record the length of time taken to reach interruption of transmission (IOT) up to a maximum of 20 years after the start of the MDA programme. We define this as the time taken for 90% of the 200 simulation runs to reach a 0% prevalence of infection, using the Imperial and SCHISTOX models [21,26]. Later, we consider the impacts of additional improvements to the drug profile by adapting the Imperial model. Here we restrict our focus to a scenario with high levels of transmission ($R_0$=1.4-4, k=0.24), with an initial baseline prevalence of >50%. For each improvement to drug efficacy, we show the mean prevalence and the mean time to reach different prevalence thresholds, including IOT (0% prevalence). We also consider alterations to the MDA whereby treatment occurs annually instead of biannually, or a small proportion of the population (the 'never treated', set at a range of values between 1–5%) are permanently excluded from MDA and never receive treatment. Although the patterns of adherence often follow a beta-binomial distribution, with those who have attended previously more likely to attend again [30,31], we use the systematic approach to demonstrate the impact of just a small proportion not adhering to treatment [27].

## Results

The effect of drug efficacy on the ability to achieve IOT within 20 years from different baseline prevalences is shown in Table 3, along with the effect of MDA coverage levels and the proportion of the population that is never treated. For both age profiles used to model *S. mansoni*, where IOT can be reached within 20 years, an increased 99% efficacy can reduce the time taken by up to four years. The benefits are smaller with *S. haematobium* but note that we assume the efficacy of PZQ is already 94% for this parasite so the increase in efficacy is smaller. Most importantly, in both cases, if the coverage with PZQ is 60%, a bigger decrease in time to IOT can be attained with an increase in coverage compared to an increase in efficacy. The largest impact on reaching IOT is the proportion of the population that are systematically never treated. In

scenarios with a high baseline prevalence (high $R_0$ value), it is impossible to reach IOT if any of the population are never treated. In moderate prevalences it is possible in some scenarios when the never treated is 1% and in low prevalence settings (small $R_0$ values but >1) it is possible but can require up to 10 years of additional MDA. If the proportion never treated is as high as 5% then it is not possible to achieve IOT for any scenario we considered.

To examine the effects of additional improvements to a potential new drug formulation, we now restrict our attention to an area of high baseline prevalence (high values of the basic reproductive number $R_0$ – $R_0$ = 3.3-4 for *S mansoni*, $R_0$ = 2 for *S haematobium*). As shown in Table 4, in this scenario the entire community must be available for treatment (0% never treated) to achieve IOT, no matter the improvements to the drug profile Table 4. If this is the case, a new drug with an efficacy against juvenile and adult worms, lasting 24 weeks can reach a 0% prevalence of *S. mansoni* faster than the current drug efficacy profile by up to four years, depending on the age intensity profile, equivalent to six rounds of MDA. The same drug would only be two years faster when used against *S. haematobium*, probably due to the already high efficacy of PZQ against this species. This drug with the combination of efficacy against juvenile worms and a slow-release period is the fastest approach to achieve IOT in this scenario but is only marginally better than a slow-release drug without

**Table 3. Number of years of MDA required for interruption of transmission when varying the baseline prevalence, MDA coverage, the proportion of the population systematically never treated and the drug efficacy. We model S. mansoni with a low and high adult burden and *S. haematobium*. MDA is annual when the baseline prevalence is low or moderate and biannual if the baseline prevalence is high.**

| | | S. mansoni, low adult burden | | | | S. mansoni, high adult burden | | | | S. haematobium | | | |
| | | 60% community coverage | | 75% community coverage | | 60% community coverage | | 75% community coverage | | 60% community coverage | | 75% community coverage | |
| Baseline prevalence | Proportion never treated | 86.3% efficacy | 99% efficacy | 86.3% efficacy | 99% efficacy | 86.3% efficacy | 99% efficacy | 86.3% efficacy | 99% efficacy | 94% efficacy | 99% efficacy | 94% efficacy | 99% efficacy |
|---|---|---|---|---|---|---|---|---|---|---|---|---|---|
| Low (8%) | 0% | 14 | 11-12 | 10 | 8 | 14-16 | 12 | 10-11 | 8-9 | 12 | 11 | 9-10 | 8-9 |
| | 1% | 18-19 | 18-19 | 17 | 16 | 20->20 | 18->20 | 18-20 | 19 | 16 | 14 | 13 | 13 |
| | 5% | >20 | >20 | >20 | >20 | >20 | >20 | >20 | >20 | >20 | >20 | >20 | >20 |
| Moderate (13%-49%) | 0% | 15-19 | 13-15 | 10-14 | 9-10 | 16-20 | 13-17 | 12-16 | 9-11 | 14-16 | 13-14 | 10-12 | 9-11 |
| | 1% | >20 | >20 | >20 | 19->20 | >20 | >20 | >20 | >20 | 18->20 | 17->20 | 19->20 | 15->20 |
| | 5% | >20 | >20 | >20 | >20 | >20 | >20 | >20 | >20 | >20 | >20 | >20 | >20 |
| High (51%-70%) | 0% | 8-15 | 7-12 | 6-10 | 5-8 | 9-15 | 7-13 | 7-11 | 5-9 | 8-11 | 8-10 | 6-9 | 6-8 |
| | 1% | >20 | >20 | >20 | >20 | >20 | >20 | >20 | >20 | >20 | >20 | >20 | >20 |
| | 5% | >20 | >20 | >20 | >20 | >20 | >20 | >20 | >20 | >20 | >20 | >20 | >20 |

**Table 4. Number of years of MDA required for interruption of transmission, starting at a high baseline prevalence (>50%). The baseline efficacy is 86.3% for *S. mansoni* and 94% for *S. haematobium*.**

| Species and infection intensity age profile | Proportion never treated | Efficacy against adults only | | Additional efficacy against juveniles: 86.3% or 94% | Slow-release period (weeks) | | | 24 week slow-release period and additional efficacy against juveniles |
| | | 86.3% or 94% | 99% | | 8 | 16 | 24 | |
|---|---|---|---|---|---|---|---|---|
| S. mansoni, low adult burden | 0% | 8-11 | 6-8 | 8-10 | 7-9 | 7-9 | 7-8 | 7-8 |
| | 1% | >20 | >20 | >20 | >20 | >20 | >20 | >20 |
| S. mansoni, high adult burden | 0% | 8-12 | 6-9 | 8-11 | 7-10 | 7-9 | 7-9 | 7-8 |
| | 1% | >20 | >20 | >20 | >20 | >20 | >20 | >20 |
| S. haematobium | 0% | 8-9 | 7-8 | 7-9 | 7-8 | 7-8 | 7-8 | 6-8 |
| | 1% | >20 | >20 | >20 | >20 | >20 | >20 | >20 |

efficacy against juveniles, or one that has a near-perfect 99% efficacy against adult worms. Little benefit is seen from a drug that has efficacy against juvenile worms without any additional improvements in the drug properties profile.

These results are broken down further for the *S. mansoni*, low adult burden case in Fig 1, showing the reduction of prevalence, the probability of achieving elimination over time and the time to reach different prevalence thresholds. This shows how the best performing drug, with 99% efficacy against adult worms, quickly reduces the prevalence in just over 6 years on average, compared with the baseline case (86.3% efficacy against adult worms), which takes just under 8 years in total. Note that results from the bar chart differ from those in Table 4, as the former was calculated using the mean, whereas the latter shows the time for 90% of simulations to reach elimination.

Interestingly, most improvements to the speed of the reduction of prevalence are realised early during MDA, when the prevalence is still high. For example, a drug with a 24-week efficacy against adult and juvenile worms reduces the prevalence from 70% to 50% in 0.8 years and 50% to 25% in 1 year, compared to the baseline case which takes 1.1 and 1.7 years to reach those intervals respectively. However, the time taken to reduce the prevalence from 10% to 1% and from 1% to 0%, takes 1.6 and 1.2 years with a 24-week efficacy period, compared to 1.8 and 1.1 years for the baseline case. The only improvement that outperforms the baseline case consistently throughout MDA is a drug with the high 99% efficacy. A 99% efficacy drug outperforms all other drug characteristics at high (>50%) and low (<10%) prevalences, but it is not as effective as a drug with efficacy lasting at least 16 weeks at the moderate prevalences (10%-50%).

If MDA is administered annually rather than biannually, as shown in Fig 2, drugs with long lasting efficacy (slow release) perform better than any other. If efficacy were to last for 24 weeks, the time taken to reach elimination would be 7 years quicker than the baseline case. In this scenario, a drug with 99% efficacy performs similarly to one which has 86% efficacy against juvenile worms, which achieve elimination approximately two years quicker than the baseline case. However, once a 10% prevalence threshold is reached, a drug with 99% efficacy is the better option, taking 4 years to achieve elimination compared to approximately 5 years in all other cases.

A similar effect is seen when the frequency is restored to biannual MDA, but community coverage is reduced to 40%, as shown in Fig 3. The time to achieve IOT is approximately 25 years but can be reduced to approximately 16 years with a drug that has efficacy lasting 24 weeks. In both cases where fewer treatments are administered, the reduction in time to reach each prevalence threshold and achieve IOT from an improved drug characteristic is greater than when coverage and frequency is high. This is also the case when the efficacy of PZQ is lower than the 86.3% we have otherwise assumed, as shown in Fig 4 where the baseline efficacy is 60%. If this were the case then the benefit of a new drug with 99% efficacy is obvious, but a slow-release formulation lasting 24 weeks would decrease the time to achieve IOT, even with the same 60% efficacy.

Fig 5 illustrates the importance of good adherence amongst the population eligible for treatment. Here, MDA is administered biannually again but 1% of the population is systemically excluded from treatment. As in Table 4, this shows that IOT is difficult to achieve when this is the case. While some drug improvements decrease the time needed to reduce prevalence to 10%, the time taken to reach elimination from a 10% or 1% prevalence is much larger than when everyone adheres to treatment during the course of MDA. Once the 10% prevalence threshold is passed, there are no benefits from improvements to treatment and the lines in Fig 5A and B converge, while the variation in bar heights in Fig 5C is very limited.

Although the above results show the potential benefits of improved drug characteristics, it is clear from comparing the baseline in Fig 1 with Figs 2-5 that delivering frequent MDA with good coverage with no proportion never treated is more important than the characteristics of the drug. To illustrate this, we compare the effect of PZQ with good coverage and assuming an efficacy of 86.3%, against an ideal new drug that has a 99% efficacy against adults and juveniles and a 24-week slow release period, with suboptimal MDA. The results in Fig 6 show that while the ideal drug is very effective at bringing down prevalence and achieving IOT within 5 years (panels A and E), when the frequency of MDA is reduced to annual administration the results are very similar to the effects of PZQ with biannual MDA (panels B and F). When

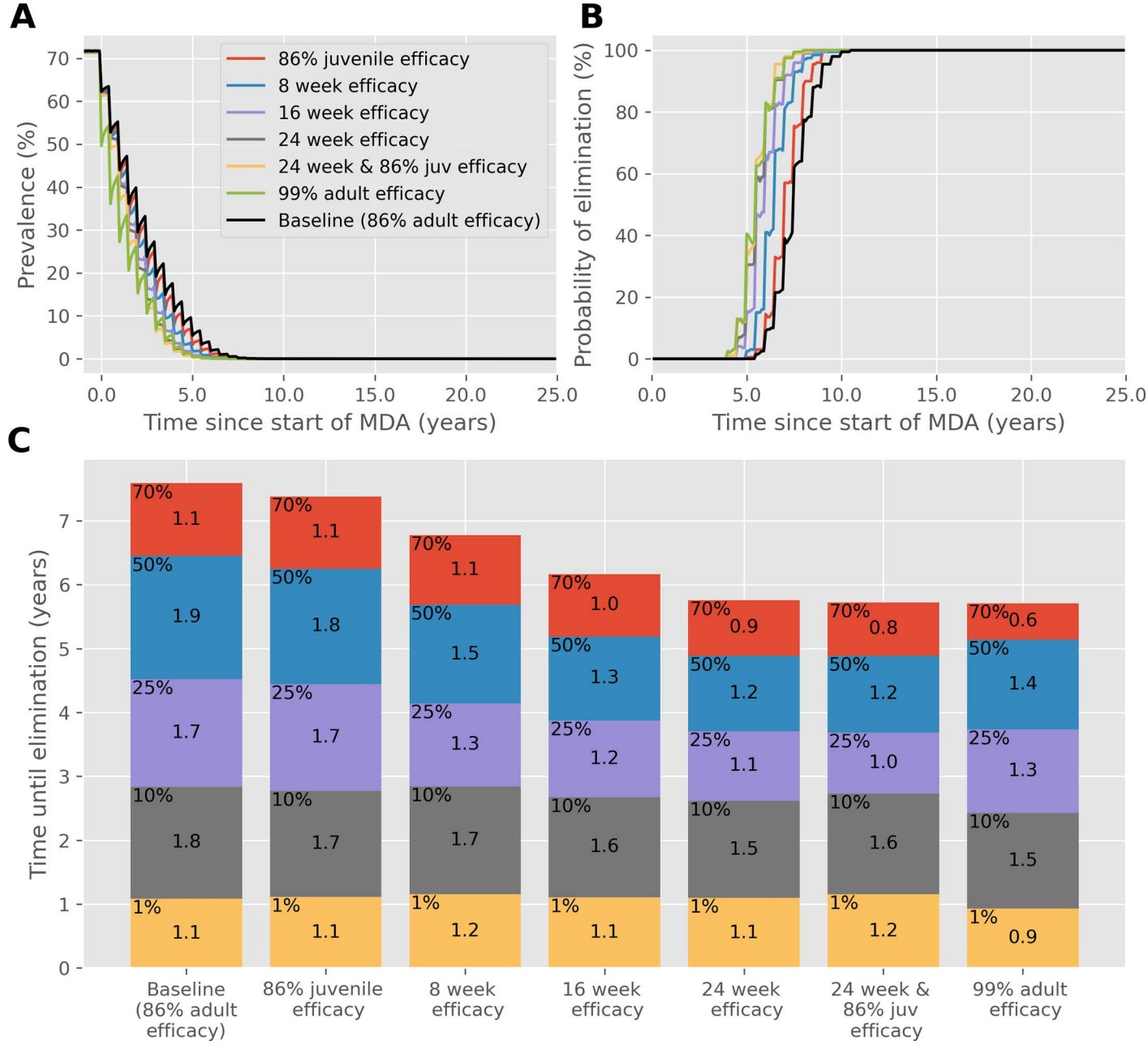

**Fig 1. Results of biannual MDA with 75% community coverage against *S. mansoni* with a low adult burden, starting with a high baseline prevalence.** (A) Prevalence, based on worm counts during MDA with a range of drug improvements over time. (B) The probability of reaching interruption of transmission with each drug improvement over time. (C) Years of MDA required to reach interruption of transmission (bar height) and the time taken to reach different prevalence thresholds (coloured blocks).

coverage is reduced to 40% (panels C and G), the ideal drug will initially reduce prevalence faster than PZQ with ideal MDA but is much slower to achieve IOT. This is also the case when 5% of the population are systematically never treated (panels D and H), where the prevalence falls much quicker with the ideal drug, but IOT is impossible to achieve within 15 years and prevalence is lower when using PZQ with ideal treatment after approximately 5 years.

PLOS Neglected Tropical Diseases

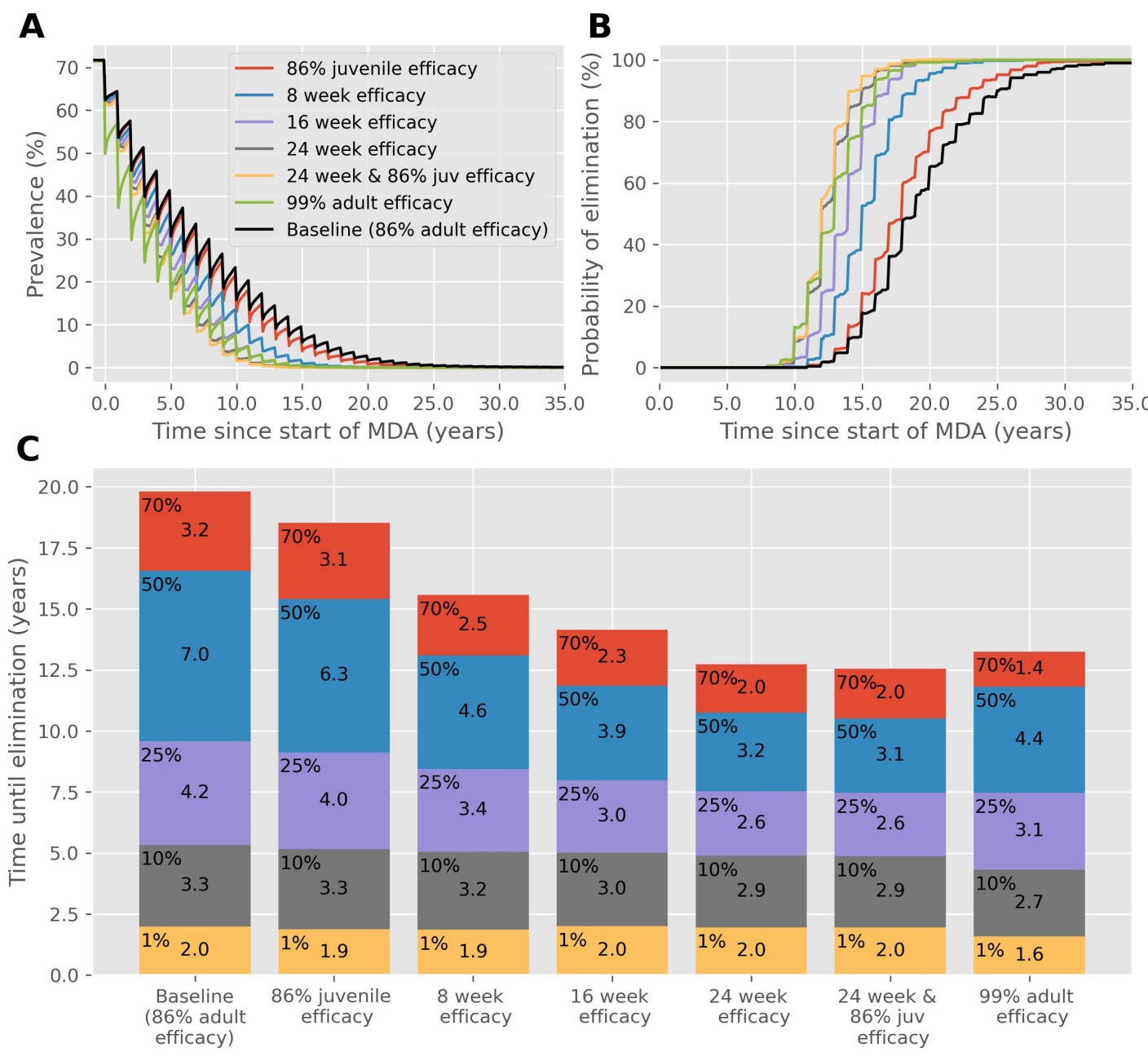

**Fig 2. As in Fig 1, except MDA is administered annually instead of biannually.** (A) Prevalence, based on worm counts during MDA with a range of drug improvements over time. (B) The probability of reaching interruption of transmission with each drug improvement over time. (C) Years of MDA required to reach interruption of transmission (bar height) and the time taken to reach different prevalence thresholds (coloured blocks).

## Discussion

Use of PZQ has long been the principal approach to reducing the prevalence of schistosome parasite infection and controlling morbidity due to high burdens of worms. However, the ambition to achieve the elimination of transmission in defined settings has led to research into new drugs, alongside other interventions such as improvements in water, sanitation and hygiene, plus snail control (the intermediate host in the life cycle) [7]. Many of these ideas have yet to be tested

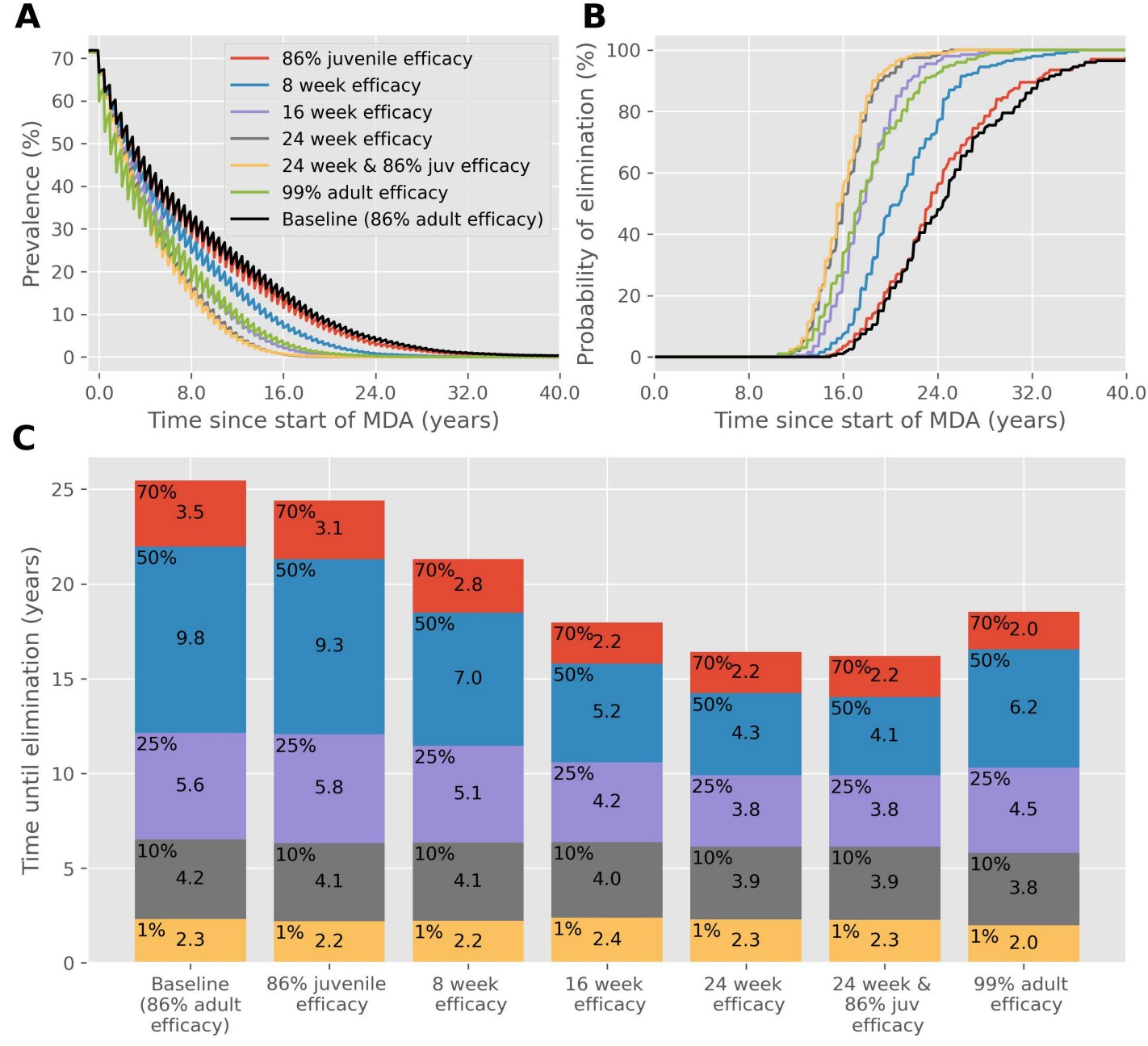

**Fig 3. As in Fig 1, except MDA coverage is 40% of the community.** (A) Prevalence, based on worm counts during MDA with a range of drug improvements over time. (B) The probability of reaching interruption of transmission with each drug improvement over time. (C) Years of MDA required to reach interruption of transmission (bar height) and the time taken to reach different prevalence thresholds (coloured blocks).

at a population level. Past discussion about what might be a sensible set of properties for any new drug to support schistosomiasis control have been based largely on what technically might be possible (e.g., activity against juvenile worms and slow-release technologies). The research described in this paper sets out to provide a more quantitative background for such discussions by investigating via simulation what the population level impact of different drug properties would be based on the predictions of a mathematical model of the transmission of *S. mansoni* and *S. haematobium* and MDA

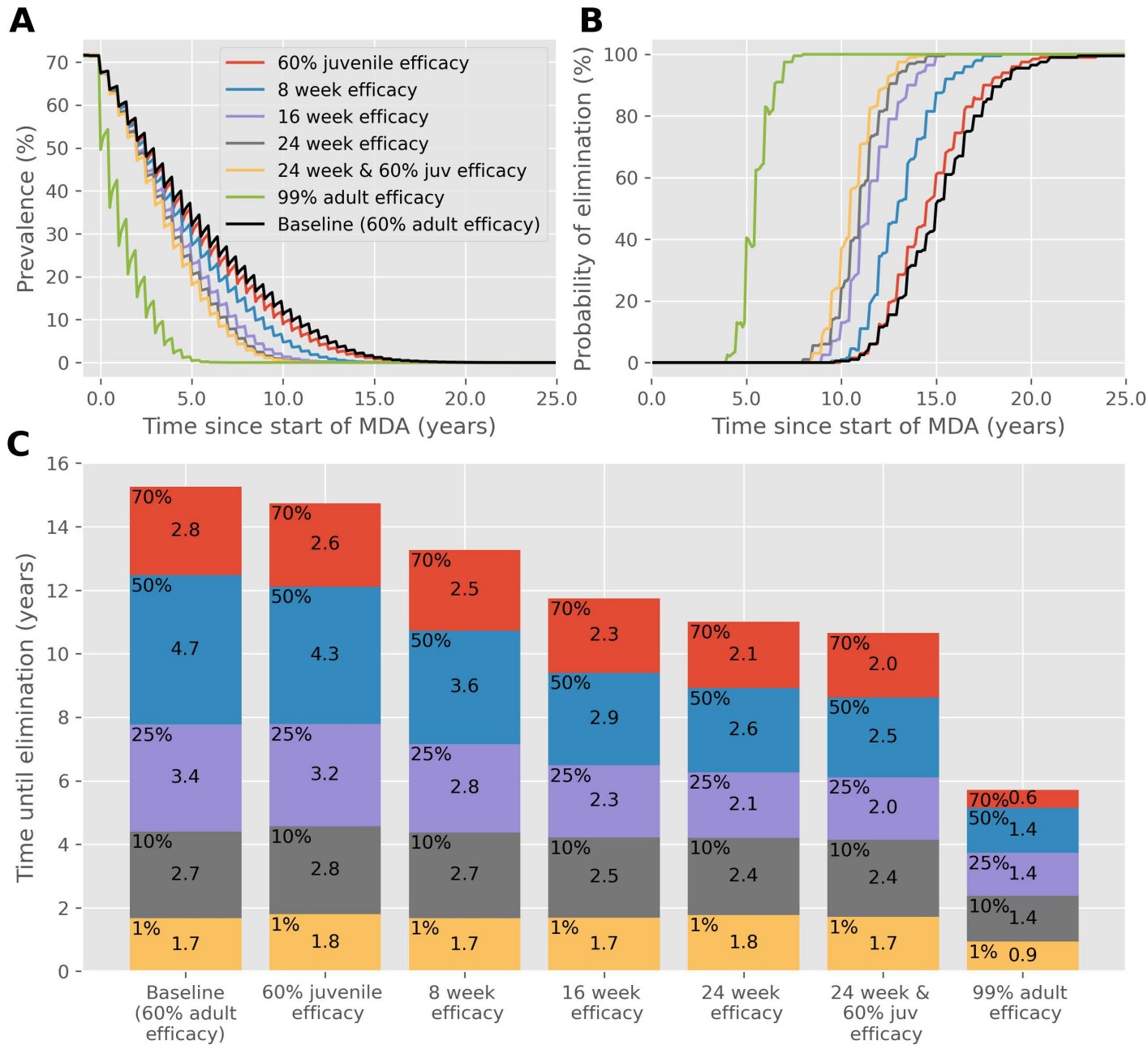

**Fig 4. As in** Fig 1**, except the efficacy of PZQ is assumed to be 60%.** (A) Prevalence, based on worm counts during MDA with a range of drug improvements over time. (B) The probability of reaching interruption of transmission with each drug improvement over time. (C) Years of MDA required to reach interruption of transmission (bar height) and the time taken to reach different prevalence thresholds (coloured blocks).

impact. The magnitude of the improvements we consider are not based on products currently in development. However, the results provide guidance on the effect that each improvement will have at the community level. The parameters of the model can be adjusted to fit a specific drug in future, when a likely candidate emerges.

Although a lack of efficacy against juvenile worms is often highlighted as a problem with PZQ, we find that at a population level this has little impact on reducing prevalence. This is due to the time it takes to mature in the human host being small

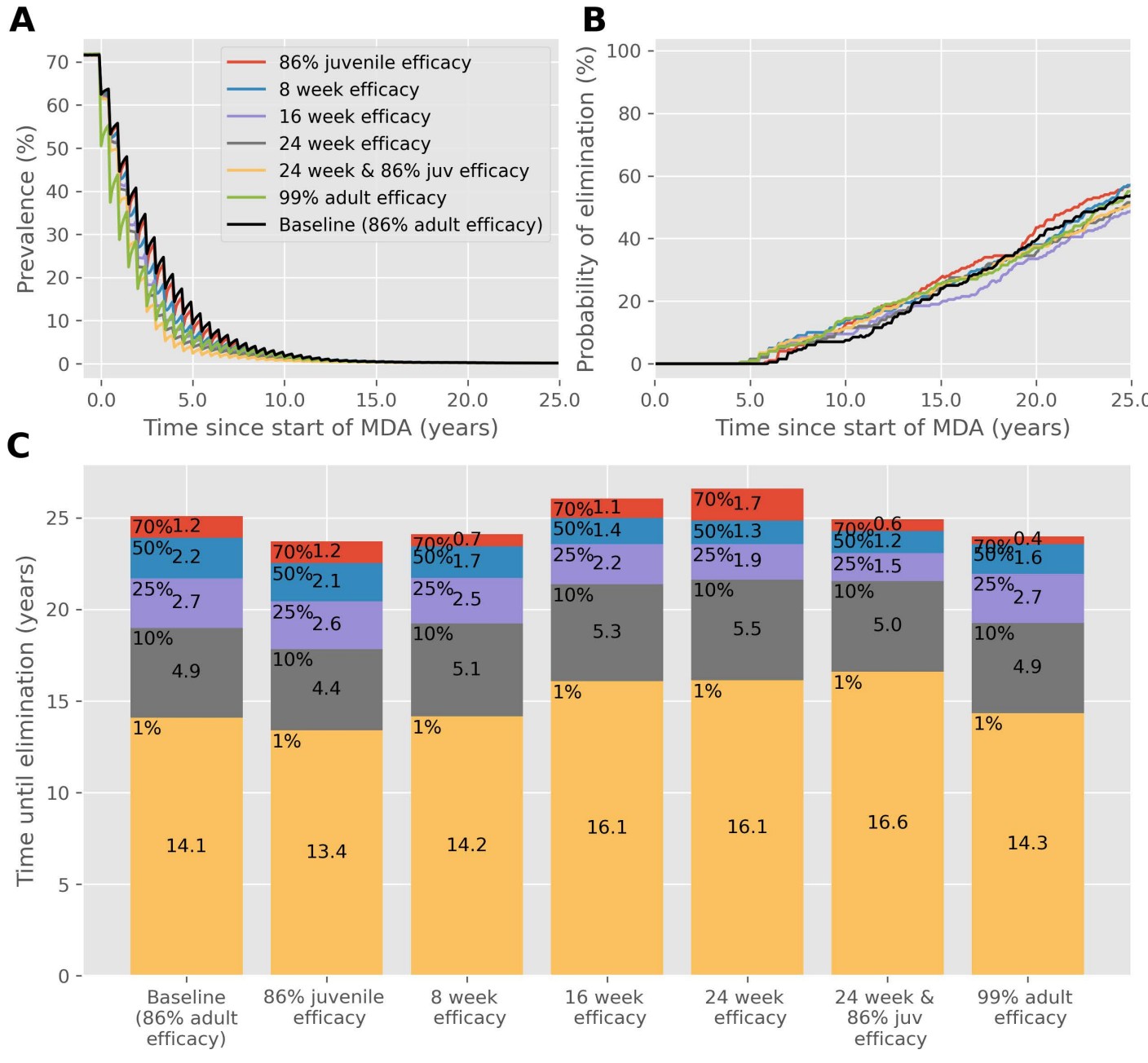

**Fig 5. As in Fig 1, except 1% of the population are systematically excluded from MDA and are therefore never treated.** (A) Prevalence, based on worm counts during MDA with a range of drug improvements over time. (B) The probability of reaching interruption of transmission with each drug improvement over time. (C) Years of MDA required to reach interruption of transmission (bar height) and the time taken to reach different prevalence thresholds (coloured blocks).

relative to adult worm life expectancy. We have assumed that the juvenile maturation period is 5 weeks on average compared to the adult life span of 5.7 years for *S. mansoni* and 4 years for *S. haematobium*. This means that the number of juveniles as a proportion of the total number of worms in the population is small, approximately 1.5-2.5%. Therefore, even the impact of a drug with 100% efficacy against juvenile worms would only have the effect of improving efficacy against all worms by 2.5% at best.

PLOS Neglected Tropical Diseases

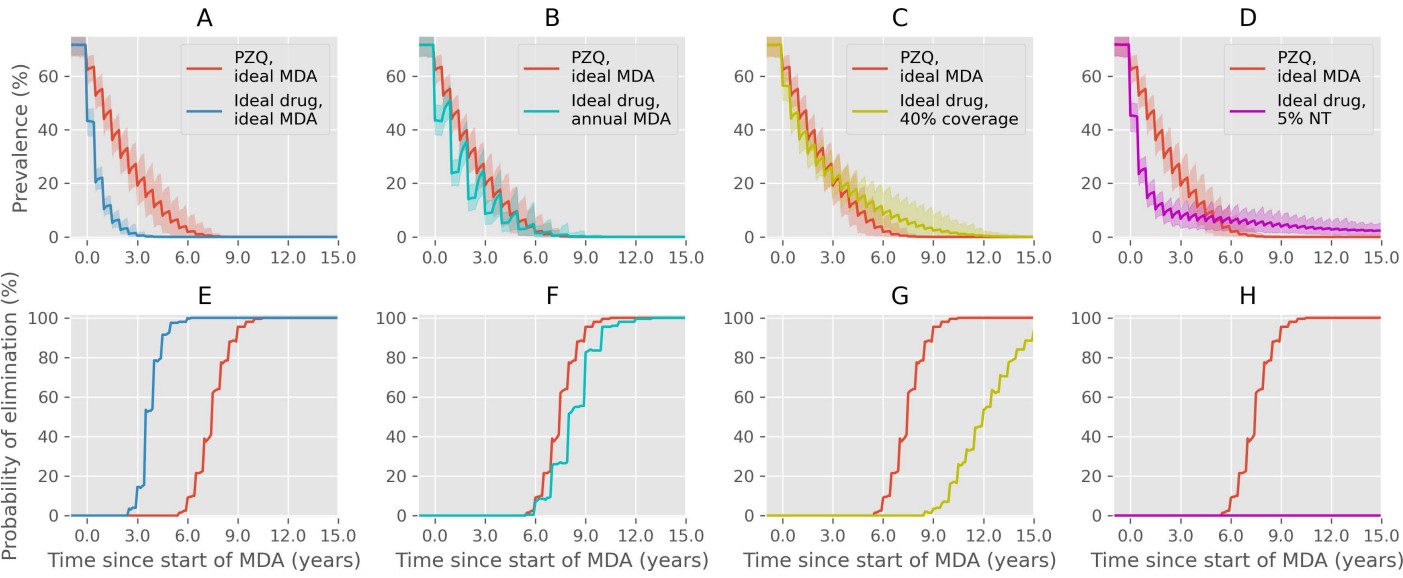

**Fig 6. A comparison between PZQ with ideal MDA, defined as 75% community coverage, biannual, 0% NT (red lines), and an ideal improved drug with 99% efficacy against juvenile and adult worms, lasting for 24 weeks.** Individual plots show prevalence (A-D) and probability of interruption of transmission (E-H) for 15 years after starting MDA. (A, E) Comparison between PZQ (red) and an ideal drug (blue), both with ideal coverage. (B, F) The ideal drug is given annually instead of biannually (cyan). (C, G) Coverage of the ideal drug is reduced to 40% (yellow). (D, H) 5% of the population are permanently excluded from MDA with the ideal drug (magenta). Shaded regions show the 95% prediction intervals from an ensemble of 200 simulation results.

It is therefore unsurprising that a drug with high efficacy against adult worms achieves better outcomes than one with an efficacy against juveniles but no improvement in efficacy against adults. A drug with 99% efficacy against adult worms is effective at decreasing the time to achieve IOT, particularly when coverage is also good. Similarly, it is unsurprising that most of the improvements don't show substantially better results for *S. haematobium* compared to the baseline (see Table 4)., because the efficacy is already quite high against this schistosome species. The estimate for the efficacy of PZQ used in this model is one that is commonly referred to [22,23], however most estimates of egg reduction rate range from 60-100% [41]. If resistance is being developed, the efficacy of PZQ in the field could be much lower. If this is the case, or if efficacy is less than assumed here, the benefits of a new drug with higher efficacy are obviously highly desirable, but a slow-release period would also be beneficial, even if efficacy remained low. The caveat to this is that a slow-release formulation may affect the way resistance develops.

What is surprising, perhaps, is the result that when a low prevalence is already reached, there appears to be no benefit to drug improvements, with the exception of a drug with a very high efficacy. This is counter-intuitive, as one might expect that there would be value in killing juveniles when approaching elimination. However, at lower prevalences, the rate of new infections will have already slowed, and thus the number of juveniles will be small, whereas there will still be many adult worms in the population particularly if not everyone eligible has received treatment, or the efficacy against adults is not sufficiently high. A drug with high efficacy against adult worms only is also the most effective at reducing prevalence from a high baseline level as almost everyone who is treated will be free from infection. However, at moderate prevalences, drugs with a long-lasting efficacy are most effective as they provide protection from reinfection and reduce the immediate bounce back in prevalence after MDA.

In this paper we mostly focus on the goal of interruption or elimination of transmission and for the purpose of the model, define this as a complete absence of worms in the population. Although this can be achieved in a closed community, there is always the potential for infection to be reintroduced and prevalence to bounce back if infected people move into or visit

areas that have cleared infection by high coverage of MDA over many years [42]. In this scenario where prevalence is low, high efficacy against adult worms and good coverage of MDA, with no proportion never treated, are more important than the other potential improvements.

The improvements in the properties of an anti-schistosome infection drug were chosen to illustrate how they might impact infection at a community level. The main conclusion arising from the simulation studies is that high coverage and high efficacy are key to success, but long-lasting efficacy can also be very effective. This is particularly the case at moderate and high prevalences, and when MDA coverage is not ideal. Activity against juvenile worms is of limited additional benefit. When coverage is low, or some of the population are systematically excluded from MDA, even an ideal drug with very high efficacy that lasts for 24 weeks is not as effective as PZQ with good coverage. For many helminth infections WHO has recommended MDA coverage targets of around 75% for school aged children. We believe this is too low. Coverage targets should be around 80–90% of the total population in moderate or high transmission settings if transmission interruption is the goal. As has already been shown elsewhere, good coverage over the course of MDA year by year, with a very low proportion of never treated, is vital [5,27]. To achieve elimination, MDA alone will not be sufficient in many cases and other interventions such as snail control will be required to achieve elimination [43]. WASH interventions are also strongly recommended as a key supplement to MDA and for long lasting general improvements in health [6].

## Author contributions

**Conceptualization:** Andreia Vasconcelos, Samuel M Thumbi, T. Déirdre Hollingsworth, Roy M Anderson.

**Formal analysis:** John Ellis, Nyamai Mutono.

**Funding acquisition:** T. Déirdre Hollingsworth.

**Methodology:** John Ellis.

**Project administration:** Andreia Vasconcelos.

**Software:** John Ellis, Nyamai Mutono.

**Supervision:** Samuel M Thumbi, T. Déirdre Hollingsworth, Roy M Anderson.

**Visualization:** John Ellis.

**Writing – original draft:** John Ellis, Nyamai Mutono.

**Writing – review & editing:** John Ellis, Nyamai Mutono, Andreia Vasconcelos, Samuel M Thumbi, T. Déirdre Hollingsworth, Roy M Anderson.

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
