## [Decision Letter · Decision Letter 0]

23 Jan 2025

How improvements to drug effectiveness impact mass drug administration for control and elimination of schistosomiasis

Dear Dr. Ellis,

Thank you for submitting your manuscript to PLOS Neglected Tropical Diseases. After careful consideration, we feel that it has merit but does not fully meet PLOS Neglected Tropical Diseases's publication criteria as it currently stands. Therefore, we invite you to submit a revised version of the manuscript that addresses the points raised during the review process.

Please submit your revised manuscript within 60 days Mar 24 2025 11:59PM. If you will need more time than this to complete your revisions, please reply to this message or contact the journal office at plosntds@plos.org. Please include the following items when submitting your revised manuscript:

We look forward to receiving your revised manuscript.

Kind regards,

Song Liang

Academic Editor

Francesca Tamarozzi

Section Editor

Shaden Kamhawi

co-Editor-in-Chief

Paul Brindley

co-Editor-in-Chief

**Journal Requirements:**

At this stage, the following Authors/Authors require contributions: John Ellis, Nyamai Mutono, Andreia Vasconcelos, Mwangi Thumbi, Deirdre Hollingsworth, and Roy Anderson. Please ensure that the full contributions of each author are acknowledged in the "Add/Edit/Remove Authors" section of our submission form.

4) We note that your Data Availability Statement is currently as follows: "All relevant data are within the manuscript.References". Please confirm at this time whether or not your submission contains all raw data required to replicate the results of your study. Authors must share the “minimal data set” for their submission. PLOS defines the minimal data set to consist of the data required to replicate all study findings reported in the article, as well as related metadata and methods (https://journals.plos.org/plosone/s/data-availability#loc-minimal-data-set-definition).

2) State what role the funders took in the study. If the funders had no role in your study, please state: "The funders had no role in study design, data collection and analysis, decision to publish, or preparation of the manuscript.".

**Reviewers' Comments:**

**Comments to the Authors:**

**Please note that one of the reviews is uploaded as an attachment.**

Reviewer's Responses to Questions

**Key Review Criteria Required for Acceptance?**

**Methods**

-Are the objectives of the study clearly articulated with a clear testable hypothesis stated?

-Is the study design appropriate to address the stated objectives?

-Is the population clearly described and appropriate for the hypothesis being tested?

-Is the sample size sufficient to ensure adequate power to address the hypothesis being tested?

-Were correct statistical analysis used to support conclusions?

-Are there concerns about ethical or regulatory requirements being met?

Reviewer #1: (No Response)

Reviewer #2: (No Response)

**Results**

-Does the analysis presented match the analysis plan?

-Are the results clearly and completely presented?

-Are the figures (Tables, Images) of sufficient quality for clarity?

Reviewer #1: (No Response)

Reviewer #2: (No Response)

**Conclusions**

-Are the conclusions supported by the data presented?

-Are the limitations of analysis clearly described?

-Do the authors discuss how these data can be helpful to advance our understanding of the topic under study?

-Is public health relevance addressed?

Reviewer #1: (No Response)

Reviewer #2: (No Response)

**Editorial and Data Presentation Modifications?**

Reviewer #1: (No Response)

Reviewer #2: (No Response)

**Summary and General Comments**

Reviewer #1: Comments attached

Reviewer #2: This manuscript explores the impact of improved drug efficacy on schistosomiasis mass drug administration (MDA) using a mathematical model. While the study has some novelty and public health relevance, the model suffers from several key methodological flaws that require significant revisions.

Major Weaknesses:

1.The model assumes perfect mixing within the population, neglecting heterogeneity in contact rates, activity patterns, and environmental exposures. This oversimplification likely biases the results and inflates the predicted effectiveness of interventions.

2.The model uses idealized drug assumptions, particularly regarding the 24-week slow-release formulation, which lacks real-world evidence. Furthermore, it assumes similar drug mechanisms for juvenile and adult worms, disregarding potential differences in drug pharmacokinetics and mechanisms of action.

3.Although the authors use two models, the validation focuses only on the consistency of main conclusions, not on thorough verification of the model’s parameterization, assumptions, and results across different scenarios. This limited validation is insufficient to address the model’s inherent limitations.

4.The model does not adequately account for real-world complexities of MDA implementation, such as poor drug adherence, population mobility, and the synergistic effects of multiple interventions. This disconnect limits the practical applicability of the findings.

Recommendation:

Given the aforementioned methodological shortcomings, I recommend a Major Revision. The authors must:

1.Thoroughly discuss the limitations of the model and clarify how these limitations may affect the results.

2.Revise or reconsider the drug assumptions to be more realistic, and perform sensitivity analysis.

3.Employ a more comprehensive validation approach to ensure the robustness of the model results.

4.Incorporate more real-world complexities in the model to improve its practical implications.

PLOS authors have the option to publish the peer review history of their article (what does this mean? ). If published, this will include your full peer review and any attached files.

**Do you want your identity to be public for this peer review?** For information about this choice, including consent withdrawal, please see our Privacy Policy .

Reviewer #1: **Yes: ** Berhanu Erko

Reviewer #2: No

**Figure resubmission:**

**Reproducibility:**



---

## [Editor Report · Decision Letter 1]

28 Apr 2025

Dear Dr. Ellis,

We are pleased to inform you that your manuscript 'How improvements to drug effectiveness impact mass drug administration for control and elimination of schistosomiasis' has been provisionally accepted for publication in PLOS Neglected Tropical Diseases.

Best regards,

Song Liang

Academic Editor

Francesca Tamarozzi

Section Editor

Shaden Kamhawi

co-Editor-in-Chief

Paul Brindley

co-Editor-in-Chief

---

## [Editor Report · Acceptance letter]

Dear Dr Ellis,

We are delighted to inform you that your manuscript, "How improvements to drug effectiveness impact mass drug administration for control and elimination of schistosomiasis," has been formally accepted for publication in PLOS Neglected Tropical Diseases.

Best regards,

Shaden Kamhawi

co-Editor-in-Chief

Paul Brindley

co-Editor-in-Chief
